# Personalized Cancer Vaccines: Current Advances and Emerging Horizons

**DOI:** 10.3390/vaccines13121231

**Published:** 2025-12-08

**Authors:** Lantian Lu, Xuehan Lu, Wei Luo

**Affiliations:** 1Department of Microbiology and Immunology, Indiana University School of Medicine, Indianapolis, IN 46202, USA; 2Frazer Institute, Translational Research Institute, Faculty of Medicine, The University of Queensland, Brisbane, QLD 4102, Australia; xuehan.lu@student.uq.edu.au; 3Indiana University Simon Comprehensive Cancer Center, Indiana University School of Medicine, Indianapolis, IN 46202, USA; 4Indiana University Cooperative Center of Excellence in Hematology (CCEH), Indiana University School of Medicine, Indianapolis, IN 46202, USA

**Keywords:** neoantigens, personalized cancer vaccines, peptide vaccines, genetic vaccines, autologous DC vaccines, vector vaccines

## Abstract

Personalized cancer vaccines represent a revolutionary frontier in oncology, harnessing the unique genetic and molecular profile of individual tumors to elicit targeted immune responses. This review provides a comprehensive overview of the current landscape and future perspectives of neoantigen-based personalized cancer vaccines, encompassing peptide, mRNA, DNA, autologous dendritic cell, and viral or bacterial vector platforms. We further discuss the integration of immune adjuvants, delivery systems, and combinational strategies, particularly with immune checkpoint inhibitions, to overcome tumor-induced immune exhaustion and improve therapeutic efficacy. Despite significant clinical progress over the past decade in this space, major challenges remain in immunogenic neoantigens prediction, streamlining individualized vaccine manufacturing, and optimization of combinational regimens to maximize durable antitumor responses. By reviewing recent preclinical and clinical studies on neoantigen-based cancer vaccines, this review highlights key advances, identifies persistent translational bottlenecks, and underscores the need for biomarker-guided mechanistically informed trials to fully unleash the clinical potential of neoantigen-based personalized cancer vaccines in the era of precision immuno-oncology.

## 1. Introduction

The development of cancer vaccines has been a long and complicated journey, shaped by several major milestones and persistent challenges. One of the greatest obstacles has been the profound heterogeneity of cancers across different patients and populations, complicating the design of universal cancer vaccines.

The successful development of preventative vaccines against infections of Hepatitis B virus (**Engerix-B**, **Recombivax HB**, and **PreHevbrio**) and Human Papillomavirus (**Cervarix**, **Gardasil**, and **Gardasil 9**) targeting oncogenic viral infections that can lead to hepatocellular carcinoma and cervical cancer, respectively, represents landmark achievements. It was reported that vaccination of Hepatitis B vaccines reduced the risk of developing liver cancer by 72% [1]. Similarly, vaccination of HPV vaccines decreased the risk of developing cervical cancer by 62% to 86% [2]. This shows that preventing the infection of oncogenic viruses could effectively decrease the risk of getting cancer. However, these successes rely on targeting viral antigens, which are fundamentally different from the highly individualized somatic mutations in most solid tumors. Thus, they do not address the core challenge of generating potent immunity against patient-specific, non-viral cancers.

Early cancer vaccine efforts historically relied on tumor-associated antigens (TAAs), many of which are expressed in multiple tissues and shared across patient but tend to be weakly immunogenic to central tolerance. These limitations contributed to decades of disappointing clinical outcomes. Hence, the progress in developing therapeutic cancer vaccines lagged for decades. Randomized phase 3 clinical trials, including those targeting follicular lymphoma, failed to meet primary endpoints and dampened enthusiasm for this field [3,4]. To date, the only FDA-approved therapeutic cancer vaccine is Sipuleucel-T [5], which involves ex vivo activation of autologous antigen-presenting cells (APCs) with a fusion protein containing prostate antigens and granulocyte-macrophage colony-stimulating factor (GM-CSF), a cytokine that stimulates APCs and recruits immune cells, to prompt immune responses against metastatic castration resistant prostate cancer.

The major obstacle of developing therapeutic cancer vaccines is the extensive heterogeneity of tumor antigens across individuals and tumor types. This challenge is further complicated by the polymorphism of human leukocyte antigen (HLA) molecules which regulate antigen presentation. Because each individual expresses a unique combination of HLA class I and II molecules, even the same tumor antigens, whether TAAs or antigens solely expressed in tumor tissues, may be presented efficiently in one patient efficiently but poorly or not at all in another. These layers of interpatient variability have made it extremely difficult to design universally effective therapeutic cancer vaccines. This has shifted increasing scientific focus toward targeting individualized antigens rather than shared antigens.

The landscape changed dramatically with the recent emergence of next-generating sequencing (NGS), immunogenomics, and improved bioinformatic pipelines. These advances enabled rapid identification of patient-specific somatic mutations and the prediction of neoantigens, peptides absent from the normal proteome. These neoantigens hold the potential to elicit robust antitumor T cell responses, with central tolerance to the neoantigen being minimized, as they originate from somatic mutations. The convergence of high throughput neoepitope prediction, vaccine delivery platforms, and therapy combination strategies overcoming immunosuppressive tumor microenvironment (TME), has propelled the development of personalized cancer vaccines (PCVs). These conceptual and technological advances have shifted cancer vaccine research from a phase of stagnation into a dynamic era of precision immuno-oncology. In this review, we summarize the major milestones, clinical outcomes, and ongoing challenges in the development of PCVs.

## 2. Neoantigens—The Game Changer

Unlike tumor-associated antigens (TAAs) that are overly expressed by tumor tissues, neoantigens are tumor-specific antigens (TSAs) resulting from various sources [6]. Neoantigens can be generated through single nucleotide variants (SNVs), base insertion-deletion (INDELS), and gene fusions at genomic levels. Alternative splicing, polyadenylation, RNA editing, and allegedly non-coding regions at transcriptomic levels can also lead to the production of neoantigens. At proteomic levels, dysregulated translation, post-translational modifications such as glycosylation or phosphorylation, aberrant proteasome processing and transporter associated antigen processing all contribute to the generation of neoantigens [7].

Upon translation, neoantigens are processed by antigen-presenting cells (APCs), such as dendritic cells (DCs), and subsequently presented on major histocompatibility complex (MHC) class I or II molecules, also known as human leukocyte antigen (HLA) I or II, for recognition by CD8^+^ or CD4^+^ T cells, respectively. While both subsets contribute to antitumor immunity, emerging evidence indicates that neoantigen-reactive CD4+ T cells can play a dominant role in orchestrating effective antitumor responses [8,9,10]. Multiple studies have reported that most of the ex vivo IFN-γ response was generated by neoantigen-reactive CD4^+^ T cells upon neoantigen vaccination, even though selection of epitopes was primarily based on HLA class I binding [9,10]. This neoantigen-specific CD4^+^ T cell immunodominance could potentially be attributed to multiple reasons, such as broader peptide-binding promiscuity of MHC-II molecules [11], less constrained priming for CD4^+^ T cells than CD8^+^ T cells (as only specific DC subsets cross-present to CD8^+^ cells [12]), and better expansion of CD4^+^ T cells upon vaccination [10].

Vaccination with neoantigens in cancer patients can both amplify pre-existing neoantigen-specific T cell responses when present and prime de novo T cell clones capable of recognizing neoantigens introduced by the vaccines [13]. Notably, most PCV-associated responses are reported to arise de novo [8,14,15], suggesting immune evasion of tumors could be mediated by eliminating neoantigen-reactive clones. The two landmark trials by Ott et al. and Sahin et al. in 2017 both revealed that the majority of neoantigen-reactive T cell responses post-vaccination came from newly expanded clonotypes from naïve T cells rather than pre-existing ones detected before vaccination [9,10].

The formulation of PCVs remains time-consuming, typically requiring several months upon tissue acquisition, irrespective of the vaccine platform. To circumvent this limitation, the concept of ‘off-the-shelf’ shared neoantigen vaccines was proposed less than a decade ago. Although relatively rare, shared neoantigens arise from recurrent mutations in tumor driver genes such as KRAS, tumor suppressor genes such as TP53, and viral oncogenes such as HPV-derived *E6* and *E7*. Therefore, shared neoantigens can be present across patients and even distinct cancer types. Goloudina et al. provided a comprehensive summary of shared neoantigens observed in cancer [16], highlighting frequent TP53 and KRAS mutations found in colorectal, lung, and pancreatic cancers, BRAF mutations in bladder, kidney, and thyroid cancers, EGFR mutations in hematological malignancies and lung cancer, IDH1 mutations in gliomas, and PIK3CA mutations in breast cancer.

Typical examples of shared neoantigens that may hold translational potentials include p53 R175, G245, R248, R249, R273 and R282 variants [17], KRAS G12D/V and G13D [16], BRAF V600 variants (E/K/A/D/G/L/M/Q/R) [18,19], EGFR hotspot alterations (exons 2–7, 19 and 21 deletions, L858R, T790M, and G719X) [16,20], HER2 S310F/Y, L755S and V777L [16], PIK3CA H1047R, E545K and E542K [21], or IDH1 R132H [22]. The HLA-restricting elements have been identified for some of these shared neoantigens listed above [16].

Peterson et al. compared personal and shared frameshift neoantigen vaccines in a mouse breast cancer model [23]. Both personal and shared neoantigen vaccines elicited robust neoantigen-specific T cell responses, showing comparable protection against tumor growth. This suggested that when high-quality shared neoantigens exist, a shared vaccine has the potential to match a personalized vaccine in generating antitumor responses.

Currently, several shared neoantigen-based cancer vaccines are being evaluated in clinical studies. **ADXS-503** (Advaxis Inc., Princeton, NJ, USA) is an attenuated listeria monocytogenes-based vector vaccine that is engineered to express 11 shared neoantigens derived from KRAS, TP53, and EGFR mutations, as well as 11 proprietary TAAs. A phase 1/2 study completed in 2022 evaluated **ADXS-503** (NCT03847519) as monotherapy or in combination with pembrolizumab in metastatic squamous or non-squamous non-small cell lung cancer (NSCLC) patients [24]. For patients that were refractory to pembrolizumab, **ADXS-503** was added within 12 weeks after documented disease progression. Eventually, this combination achieved an objective response rate (ORR) of 15.4% (2/13) and a disease control (DCR) of 46.2% (6/13). In another cohort where ADXS-503 plus pembrolizumab were used as first-line therapy, a DCR of 66.7% (2/3) was achieved. This study demonstrated the feasibility of cancer vaccines targeting both shared neoantigens and TAAs in controlling disease progression.

**NOUS-209** [25], developed by Nouscom (Basel, Switzerland), is built on a heterologous prime-boost viral vector platform that encodes 209 shared frameshift peptides derived from both sporadic and hereditary microsatellite instability tumors. It is now being tested in a phase 1b/2 study (NCT04041310) for cancer interception in Lynch syndrome carriers [26]. **SLATE-001**, developed by Gritstone Bio (San Francisco, CA, USA), is another shared neoantigen vaccine employing a chimpanzee adenovirus vector prime followed by a self-amplifying mRNA boost [27]. This construct originally delivers up to 20 shared neoantigens derived from both KRAS and TP53 mutations [27]. However, the T cell responses generated against shared neoantigens were biased towards HLA-matched TP53 neoantigens in the vaccines in relation to KRAS neoantigens, potentially hindering its therapeutic effects. Currently, the next generation of SLATE targeting only 4 KRAS mutations is being evaluated in combination with nivolumab and ipilimumab in advanced or metastatic NSCLC, microsatellite stable colorectal cancer, pancreatic cancer, and shared neoantigen-positive tumors in a phase 2 study (NCT03953235).

Although the concept of “off-the-shelf” neoantigen vaccines has shown promise in preclinical studies [25], it remains uncertain whether exposing the immune system to a large number of patient non-specific epitopes would dilute or constrain the expansion of truly protective neoantigen-reactive T cell clones. Addressing this question will be essential for determining the translational potential of shared neoantigens in public cancer vaccines. Shared neoantigens simplify manufacturing but risk lower specificity and higher potential for tumor immune escape. Individualized vaccines incorporating multiple patient-specific neoantigens maximize precision and TCR repertoire depth but require longer production timelines. The choice between these strategies depends on tumor type, mutational burden, and logistical constraints.

## 3. Identification of Neoantigens—The Cornerstone

The accurate identification of neoantigen is the cornerstone of PCV development (Figure 1). Recent advances in NGS including whole-exome sequencing (WES), whole-genome sequencing (WGS), and RNA sequencing, have enabled high-resolution profiling of somatic mutations and transcriptomic changes from tumor and liquid biopsies. Computational pipelines integrate such data to predict putative neoantigens based on parameters such as peptide-MHC binding affinity, proteasomal cleavage, and HLA allele specificity.

Most prediction tools rely on machine learning, especially artificial neural networks (ANNs), trained on peptide-HLA binding data and mass spectrometry-derived immunopeptidomes, enabling the capture of non-linear interactions between peptides and MHC molecules. NetMHCpan remains the most widely used predictor due to its broad HLA coverage and robust benchmarking [28,29]. NetMHCpan-4.1 is trained on more than 850,000 quantitative binding affinities and eluted ligand peptides, covering over 170 MHC molecules across multiple species, and even allows predictions for custom MHC molecules. MHCflurry has also gained broad application, integrating separate models of MHC class I binding and antigen processing, with performance validated against mass spectrometry-identified ligands [30,31].

More recent pipelines extend beyond binding-centric predictions to adopt multi-omics and multi-feature approaches. For example, ScanNeo2 incorporates canonical and non-canonical splicing, gene fusions, and other variant types to broaden the scope of candidate epitopes [32], while nextNEOpi integrates mutation, expression, and patient-specific features to estimate tumor immunogenicity and potential therapeutic responsiveness [33]. Collectively, these tools reflect a shift towards integrative frameworks that incorporate antigen processing, peptide-HLA stability, and, increasingly, TCR recognition.

Despite these advances, prediction accuracy remains a key bottleneck. The major limitations in neoantigen prediction stem from inaccurate modeling of peptide-MHC binding, incomplete characterization of antigen processing, insufficient integration of post-translational events, and the inability to predict TCR recognition. Many current pipelines rely on neural networks trained on limited datasets of eluted ligands, overlooking the diversity of HLA alleles or proteasomal cleavage rules. Recent models using artificial intelligence (AI) models such as transformer-based architectures trained on numerous peptide-HLA interaction measurements, improve prediction of stability rather than affinity alone [34]. Additionally, multi-omics workflows that integrate genomics, transcriptomics and immunopeptidomics such as ScanNeo2 were reported to reduce false-positive rates by accounting for RNA expression, allele-specific presentation, and neoepitope abundance [32,33].

As only a fraction of neoantigens is immunogenic in vivo, validation of their immunogenicity is of essential importance to guide future vaccination. Mass spectrometry-based immunopeptidomics can identify human leukocyte antigen (HLA)-bound peptides, revealing neo-epitopes presented by given HLA types. As most identified neoantigens elicit de novo responses in clinical studies [8,14], the identification of truly immunogenic neoantigens is particularly critical post-vaccination. Functional assays such as enzyme-linked immunospot (ELISpot) or flow cytometry can identify T cells that are responsive to neo-epitopes by detecting their secretion of inflammatory cytokines such as IFN-γ and TNF. In parallel, TCR sequencing can also reveal the changes in T cell clones during vaccination, reflecting the expansion and activation of T cell populations upon neoantigen vaccination; however, this only provides indirect evidence to responses to neoantigens, as TCR specificity for neoantigens is not directly resolved. Additionally, neoantigen-loaded MHC tetramers have been used in several studies to identify neoantigen-reactive T cells, which in turns validate the immunogenicity of these neoantigens in activating T cells [35,36,37].

Genocea Biosciences (Cambridge, MA, USA) identified inhibitory neoantigens termed as “inhibigens” that abolished antitumoral responses of other neoantigens in a murine model [38]. Studies by Lam et al. suggested that those “inhibigens” are unlikely to be Treg epitopes [39], and the mechanisms of how “inhibigens” downregulate T cell responses remains unclear. Similarly, EpiVax Oncology (Providence, RI, USA) has also discovered another category of inhibitory neoantigens that promote immune tolerance through the expansion and action of Tregs [40]. These findings highlighted the importance of careful neoantigen selection, as the antitumoral immune responses generated by a pool of neoantigens have the risk of being sabotaged or even overturned by inhibitory neoantigens.

The sensitivity of current in silico neoantigen prediction tools remains suboptimal. Most studies report fewer than 60% of predicted neoantigens elicit measurable immune responses in vivo or ex vivo following vaccination in animal models or patients. In early preclinical work, Kreiter et al. reported that fewer than 10% of 184 predicted neoantigens derived from three murine tumor lines induced detectable CD8^+^ T cell responses, suggesting the high false-positive rate inherent to prediction pipelines [41]. Human data seemed to mirror this observation. For example, Zhang et al. found that only 31.1% (14 of 45) of selected neoantigens trigger Th1 cytokine secretion detected by flow cytometry [37]. Other investigators reported similar trends, showing that less than half of the vaccine-encoded epitopes generated T-cell responses after immunization. In one autologous DC vaccine study, responses were detected to only 42% (14 of 33) of predicted peptides [42]. Similarly, Awad et al. reported that 55% of epitopes in their study generated CD4^+^ and/or CD8^+^ responses across all 13 patients evaluated [8]. However, as not every clinical report provides corresponding per peptide immunogenicity rate, it is difficult to draw a conclusion that only a small fraction of candidate neoantigens screened by predictions can trigger neoantigen-specific responses. In addition, post-in vitro stimulation ELISpot has demonstrated better sensitivity in detecting truly immunogenic neoantigens in patients, suggesting the low frequency of neoantigen-reactive clones [15]. Collectively, these findings indicated the limited predictive accuracy of current algorithms and underscore the need to improve the sensitivity and specificity of in silico tools for neoantigen identification to enhance vaccine design.

## 4. Current Neoantigen Vaccine Platforms

To date, neoantigen vaccines have been studied across a broad spectrum of delivery platforms in clinical studies, including synthetic peptides, pulsed dendritic cells, nucleic acids (DNA and mRNA), and viral or bacterial vectors (Figure 2). Among these, peptide- and mRNA-based platforms have emerged as the most widely adopted approaches for delivering patient-specific neoantigens in the treatment of solid tumors (Table 1).

Peptide-based vaccines, particularly synthetic long peptides (containing both CD4^+^ and CD8^+^ T cell epitopes), are favored for their simplicity of design. However, they face several limitations, such as low immunogenicity without the help of molecular adjuvants, as well as manufacturing scalability. Additionally, because hydrophobic peptides are difficult to synthesize through solid-phase peptide synthesis, potentially immunogenic neo-epitopes may be excluded from vaccine design if they are highly hydrophobic [43]. However, the use of dimethyl sulfoxide at a low concentration helps with solubilizing peptides that are poorly water soluble [8,14]. Other novel delivery strategies have also been employed to deliver peptide neoantigens. Feola et al. used charge-modified neoantigen peptides to co-deliver with oncolytic viruses [44]. A poly-lysine tail was added to different neoantigens to ensure the positive charge, so that the peptides can be physically complexed with negatively charged oncolytic viruses through electrostatic interactions. Currently, this technology (PeptiCRAd) is under clinical evaluation with TAAs such as NY-ESO-1- or MAGE-A3-derived peptides instead of neoantigens (EudraCT 2021-002529-13).

By contrast, mRNA vaccines have emerged as a flexible and scalable platform that can encode many more patient-specific epitopes in a single construct. Further, mRNA molecules display intrinsic adjuvanticity and can be rapidly manufactured. The use of Moderna’s **mRNA-4157** in high-risk melanoma patients showcased the translational promise of this platform, showing a 44% reduction in tumor recurrence when **mRNA-4157** was combined with pembrolizumab, compared to pembrolizumab monotherapy [45]. Currently, **mRNA-4157** has been exclusively combined with pembrolizumab for clinical evaluations [46,47]. Unlike **mRNA-4157** which was delivered to patients intramuscularly [45], **Autogene cevumeran** developed by BioNTech is given to patients through intravenous administration to target DCs residing in the lymphoid compartments [15]. BioNTech’s techniques utilized unmodified mRNA to complex with cationic liposomes through electrostatic interactions to broadly target lymphoid dendritic cells without ligand decoration, while Moderna’s techniques use modified uridine (N1-methyl-pseudouridine) [48] in mRNA for encapsulation in lipid nanoparticles (LNPs) to reduce innate sensing.

Like mRNA vaccines, DNA vaccines benefit from the polyepitope vaccine design, since a single DNA construct can accommodate more than a dozen neoepitopes. Another distinct advantage of nucleic acid-based vaccines is that immune-stimulating agents can be encoded with antigens in the same construct, ensuring the co-delivery of antigens and adjuvants into the same cells. Szymura et al. reported a DNA neoantigen vaccine design encoding LPL-derived Ig single chain variable fragments as neoantigens and CCL20 [49], a chemokine that recruits immune cells such as dendritic cells, macrophages, B cells and T cells. Similarly, CCL3, another chemoattractant, was encoded in a DNA vaccine design to target antigen-presenting cells (APCs), thereby enhancing APC maturation and recruiting other cell subsets [50]. Plasmids encoding the inflammatory cytokines such as IL-12 have also been employed as adjuvants to co-formulate with neoantigen DNA vaccines [51]. Additionally, the development of electroporation devices during the past few decades has advanced the approach of DNA vaccine delivery, greatly enhancing cellular uptake of plasmid DNA, antigen expression, and DNA immunogenicity [37,51].

Neoantigen-pulsed autologous DCs have been employed as a robust platform to train patients’ T cells to recognize neoantigens. Both neoantigen peptides [52,53,54,55] and mRNA [42] have been used to load DCs. This approach ensures the successful loading of neoantigens to autologous DCs, as well as DC maturation, thereby promoting the interactions between T cells and APCs to generate neoantigen-specific antitumor responses. The downside of this approach is that the needle-to-needle turnaround is longer than other approaches [42,52], as (1) neoantigen peptides or mRNA must be prepared prior to stimulation of autologous DCs; (2) DCs must be isolated from peripheral blood mononuclear cells (PBMCs) and cultured for several days prior to neoantigen stimulation and cytokine stimulation; (3) upon stimulation, matured DCs must undergo quality control before reinfusion, during which qualified pulsed DC populations are expected to produce cytokines such as IL-12 [52] or IP10 [42].

Emerging evidence has suggested that certain bacteria can naturally home to tumors [56] and modulate antitumor immunity [57,58], providing a rationale for the development of therapeutic bacterial vector vaccines against solid tumors. Hecht et al. reported the first neoantigen-targeted bacterial vaccine, **ADXS-NEO**, an attenuated live Listeria monocytogenes-based vector engineered to secrete a fusion protein composed of multiple personalized neoantigens and a truncated fragment of listeriolysin O serving as an adjuvant. Although a phase I study in 2019 confirmed the safety and immunogenicity of **ADXS-NEO** [59], the program was subsequently discontinued due to strategic and business considerations rather than scientific or safety concerns. More recently, Redenti et al. designed a probiotic Escherichia coli Nissle 1917-based vector to deliver neoantigens in mice [60]. Incorporation of listeriolysin O facilitated cytosolic delivery of neoantigens and enhanced cross-presentation by MHC I molecules, leading to potent CD8+ T cell responses in murine melanoma models. These studies highlighted the translational potential of bacterial vector vaccines for neoantigen-based cancer immunotherapy.

Compared to bacterial vectors, viral vectors have demonstrated more advanced clinical translation and manufacturing feasibility for PCVs. For this approach, a heterologous prime-boost strategy is often employed to avoid generating redundant vector-specific responses, ensuring the successful expansion of neoantigen-specific T cells. **NOUS-PEV** (Nouscom), like **NOUS-209**, employs a great ape adenovirus (GAd)/Modified Vaccinia Ankara (MVA) heterologous system and is evaluated in a phase 1 study [61]. Similarly, **GRANITE** (Gritstone Bio) utilizes a chimpanzee adenoviral vector for priming, but it uses self-amplifying mRNA to boost [62]. Currently, **GRANITE** is being evaluated in a phase 2 study. The turnaround time of manufacturing neoantigens-loaded viral vaccines varies amongst different systems. It was reported that the average turnaround time for the GAd was around 55 days, while that for MVA was 81 days [61].

## 5. Clinical Overview of PCVs

During the early era of the development of neoantigen-based PCVs, two pioneering studies by Ott et al. and Sahin et al. showed that personalized PCVs in combination with ICIs could induce the generation of neoantigen-specific CD4^+^ and CD8^+^ T cells [9,10]. Although pre-existing neoantigen-specific T cell responses were found to be amplified post-vaccination, most of the responses were reported to be de novo. This is in line with the immune evasion mechanism of cancer by creating an immunosuppressive TME through downregulating T cell clones that confer protection against malignances. Hence, PCVs showed the potential to enhance the recognition of protective neoantigens by the immune system.

Initial clinical investigations of PCVs primarily enrolled patients with advanced and/or metastatic disease who had exhausted conventional treatment options [9,10,14,63,64]. Although vaccination in refractory population occasionally lead to disease stabilization or delayed progression, the profound immunosuppression and high tumor burden characteristics of late-stage disease often limited vaccine efficacy. These observations underscored the importance of disease setting, as PCVs may achieve greater therapeutic effect when implemented earlier, before extensive immunoediting and T-cell dysfunction occur. Reflecting on this rationale, recent studies have shifted toward evaluating PCVs as part of first-line or adjuvant therapy regimens. For example, early data from the NOUS-PEV vaccine combined with pembrolizumab as first line therapy in metastatic melanoma patients showed encouraging antitumor activity, with one patient achieving complete response, three with partial response, and one maintaining stable disease at follow-up. Similar early-line trials are ongoing across tumor types, aiming to test whether vaccination can synergize with ICIs to augment response durability and minimize relapse.

In the current clinical landscape, PCVs are increasingly employed as adjuvant therapies following standard-of-care therapies including surgical resection, chemotherapy, radiotherapy, targeted therapy, or immunotherapy, to prevent disease recurrence (Table 2). These studies reinforce the potential of PCVs to consolidate tumor regression and maintain long-term immune surveillance in minimal disease settings, paving the way for integrating precision medicine into multimodal oncologic therapy.

## 6. Immune Adjuvants: Unmet Clinical Needs

Due to the intrinsic low immunogenicity of peptide antigens, peptide PCVs use adjuvants to promote their immunogenicity. Amongst the commonly used adjuvants for peptide PCVs, poly-ICLC, a TLR3 agonist, remains popular in clinical studies [67]. This adjuvant interacts with TLR3 and MAD5, leading to APC maturation and promoting their migration to the draining lymph nodes [76,77]. Blass et al. used a multi-adjuvant strategy combining poly-ICLC and Montanide, an emulsion-based adjuvant, in synthetic long peptide neoantigen vaccines [66]. This approach together with local and systemic ICIs induced a broad and diverse repertoire of vaccine-specific T cells in both peripheral blood and at the tumor site. Another popular adjuvant used for peptide PCVs is Granulocyte-Macrophage Colony-Stimulating Factor (GM-CSF), a cytokine that recruits myeloid cells and prompt their differentiation into DCs. Instead of co-formulation with peptide PCVs, GM-CSF is often administered to cancer patients prior to peptide neoantigen vaccination [43,68,69].

Although not yet tested in human PCVs in clinical studies other TLR agonists such as TLR7/8 agonists and TLR9 agonists, as well as Stimulator of Interferon Genes (STING) agonists, have been widely evaluated in the pre-clinical settings. Castro Eiro et al. utilized a dual adjuvant system incorporating TLR9 agonist K3 CpG and STING agonist c-di-AMP, for neoantigen peptide vaccines in a murine melanoma model [78]. This combination outperformed Addavax, and poly (I:C) in generating antigen-specific T cell responses. In addition, combination of K3 CpG and c-di-AMP overcame the Th2 skewed responses induced by c-di-AMP alone, with a decrease in the number of IL-5-secreting splenocytes. More importantly, combination of K3 CpG and c-di-AMP showed better antitumor synergies with anti-PD-1 therapy, compared with Addavax, in inhibiting tumor growth.

Lynn et al. conjugated charged modified neoantigen peptides with agonists of TLR2/6, TLR7/8, TLR9 or STING [79]. These amphiphilic structures self-assemble in aqueous solution, forming nanoparticles with a size less than 50 nm. Interestingly, all conjugates except for peptide-TLR2/6a conjugates induced significantly stronger neoantigen-specific CD8^+^ T cell responses compared to naïve control. However, such strategy requires extensive conjugation of neoantigen peptides with TLR agonists and purification of the conjugates for PCVs where multiple peptides are delivered to a patient, limiting its manufacturing scalability and clinical translation.

Wu et al. employed an in situ vaccination approach combining low-dose cisplatin with TLR7/8/9 agonists. The use of chemotherapy and multiple TLR agonists (R848 and CpG1018) outperformed single modality alone in inhibiting tumor growth in a breast cancer model [80]. Abscopal effect on distant tumors was also observed after combination therapy. Interestingly, more distinct tertiary lymphatic structures near the tumor were induced by this combination therapy, compared to single modality alone, suggesting improved coordination between innate and adaptive immunity within the tumor microenvironment. These regimens highlight the potential of combinational adjuvants in peptide PCVs to generate robust antigen-specific T cell responses to inhibit tumor growth and metastases.

One limitation of current clinical studies on peptide PCVs is the lack of direct comparison between different adjuvants within the same trial. Several factors have been found to contribute to this. First, clinically licensed vaccine adjuvants remain limited, including only aluminum salts, MF59, AS01, AS03, AS04, and CpG1018 [81], while others frequently used in vaccine trials, such as poly-ICLC and MPLA, are still considered investigational. Moreover, some of the licensed adjuvants such as aluminum salts induce a Th2-skewed response, whereas a Th1-skewed response is more favorable for antitumor immunity, further limiting the options of adjuvants for PCVs. Second, in early-phase trials where safety and proof of concept are the primary endpoints, systemic testing of multiple adjuvants is often not prioritized. Moving forward, identifying and validating effective adjuvant partners for PCVs, particularly peptide PCVs, will be essential to fully unleash their therapeutic potentials.

In contrast to peptide formulations, nucleic acid-based PCVs possess intrinsic immunostimulatory properties, as the nucleic acids themselves can engage pattern-recognition receptors and trigger innate immune activation. Specifically, RNA molecules can interact with endosomal TLRs such as TLR3, TLR7, and TLR8, as well as cytosolic sensors including RIG-I and MDA5, leading to the induction of type I interferons and other proinflammatory cytokines that support antigen presentation and T cell priming [82]. Nevertheless, the unprotected nucleic acids are prone to rapid degradation in vivo and may elicit uncontrolled inflammatory signaling if not properly formulated.

To achieve efficient delivery and optimal immune activation for mRNA PCVs, liposomes or LNPs have been employed as a delivery system. Autogene cevumeran utilizes an RNA-lipoplex system as aforementioned, leveraging mRNA’s intrinsic properties to activate innate sensing receptors to trigger TLR7/8 and type I IFN response. mRNA-4157 uses modified mRNA and encapsulates the mRNA molecules within LNPs to reduce TLR activation. However, LNPs themselves can act as a potent immunoadjuvant, as lipid components can activate inflammasome or TLR-dependent signaling [47].

In summary, adjuvants play a pivotal role for the efficacy of PCVs. While peptide PCVs rely heavily on exogenous adjuvants such as poly-ICLC, GM-CSF, and merging TLR/STING agonists to drive potent cellular immunity, nucleic acid-based PCVs inherently engage innate immune sensors through their molecular backbones or delivery vehicles. The expanding understanding of adjuvants’ modes of actions, as well as the development of novel immune adjuvants will be instrumental to guide the development of the next generation PCVs toward enhanced immunogenicity and translational feasibility.

## 7. Combinational Therapy: Further Unleashing PCV’s Potential

Tumor-infiltrating neoantigen-reactive T cells commonly exhibit high expression of exhaustion markers such as programmed cell death protein 1 (PD-1), T-cell immunoglobulin and mucin-domain containing-3 (TIM-3), T-cell immunoglobulin and ITIM domains (TIGIT), and Lymphocyte-activation gene 3 (LAG-3), due to the chronic and persistent antigen stimulation they experience within the immunosuppressive tumor microenvironment (TME). This exhaustion state diminishes the durability of tumor-reactive immune responses, representing a substantial hurdle for cancer immunotherapy.

Over the past few years, combination therapies have been increasingly seen as an important approach to improve the clinical outcome of personalized neoantigen vaccination. The majority of clinical studies on neoantigen-based cancer vaccines have combined the use of immune checkpoint inhibitors (ICIs) such as anti-CTLA-4 (e.g., Ipilimumab) [66], anti-PD-L1 (e.g., Atezolizumab) [15,72,74], anti-PD-1 (e.g., Pembrolizumab and Nivolumab) [45,46,59,75], or a combination [66]. This approach helps re-invigorate tumor-infiltrative cytotoxic T cells and maintains their effector functions where immunosuppression presents. Combinational use of ICIs and neoantigen vaccines in murine models have shown synergistic effects in promoting higher infiltration of CD8^+^ T resident memory cells into tumors, contributing to stronger antitumor responses [83]. Weber et al. reported that anti-PD-1 (Pembrolizumab) when in combination with mRNA neoantigen vaccines reduced 44% recurrence risk in patients with resected melanoma, compared to anti-PD-1 monotherapy alone [45].

D’Alise et al. used viral vector PCV in combination with pembrolizumab as first line therapy in metastatic melanoma patients. Neoantigen-specific responses were observed in all patients receiving both the prime and boost vaccination. The best overall responses observed in this study were one complete response, three partial responses and one stable disease. Three patients predicted to be responders to pembrolizumab based on an IFN-γ/IMS ratio higher than -1 showed either complete response or partial response. Interestingly, two patients predicted to be non-responders to pembrolizumab showed partial response or stable disease upon receiving the combinational therapy, suggesting the effectiveness of PCV in improving therapeutic effect of ICI.

Blass et al. modified their peptide PCV by incorporating Montanide in addition to poly-ICLC, and administering ipilimumab locally at the vaccination site [66]. This new formulation (NeoVaxMI) increased the proportion of neoantigens eliciting ex vivo IFN-γ ELISpot responses, compared with their earlier NeoVax formulation using poly-ICLC-adjuvanted neoantigens alone (64% vs. 19%). However, the relative contributions of Montanide and local CTLA-4 blockade to this enhancement remain to be determined.

Ding et al. evaluated neoantigen peptide-pulsed autologous DC vaccines in 12 patients with advanced lung cancer [52]. Five patients received concurrent ICI therapy along with neoantigen vaccination (three received nivolumab, one received both pembrolizumab and ipilimumab, and one received pembrolizumab). Patients receiving concurrent ICI and neoantigen vaccines displayed better PFS and OS. However, the limited sample size precludes definitive conclusions.

In a small phase 1 study targeting late staged renal cell carcinoma, Braun et al. concluded that peptide PCV alone was highly immunogenic, and adding low-dose local ipilimumab did not significantly alter clinical outcomes [67]. Similarly, atezolizumab did not clearly enhance T cell induction or objective responses in a small heterogeneous population [15]. These findings suggested that comparisons amongst a relatively homogenous patient cohort with comparable prior treatment are required to better define the clinical benefit of adding ICIs to neoantigen PCVs.

Chemotherapies, radiations, and ablations have also been combined with neoantigen vaccinations with/without ICIs in both clinical and pre-clinical settings [43,63,70]. These modalities have the potential of enhancing neoantigen-specific immune responses. They damage tumor tissues, releasing tumor-specific neoantigens, which may allow the immune system to be primed with such neoantigens. This can then help the immune system generate stronger neoantigen-specific T cell responses upon re-encountering these neoantigens through subsequent vaccination. In addition, low dose chemotherapeutics such as cyclophosphamide can deplete immunosuppressive cells such as Tregs and myeloid-derived suppressor cells, thereby promoting vaccine-induced T cell expansion [84].

Awad et al. combined neoantigen vaccine **NEO-PV-01** with chemotherapy (pemetrexed and carboplatin) and anti-PD-1 (pembrolizumab) as first-line therapy for non-squamous non-small cell lung cancer [8]. Robust antigen-specific de novo T cell responses were generated in all patients receiving **NEO-PV-01**. Despite receiving an immunosuppressive chemotherapy prior to vaccination, the immunogenicity of **NEO-PV-01** combined with pembrolizumab was not significantly changed, compared to the prior study where only **NEO-PV-01** and pembrolizumab were scheduled [14].

Rojas et al. investigated the feasibility of Autogene cevumeran that encodes up to 20 patient-specific neoantigens, with ICI after tumor resection in pancreatic ductal adenocarcinoma patients as adjuvant therapy, prior to chemotherapy [72]. Among the 16 patients who received the PCV, half (8/16) mounted vaccine-specific T cell responses and did not show recurrence at the 18-month median follow-up. However, the contribution of subsequent chemotherapy to clinical outcome or immunological modulation in this regimen was not fully addressed.

A retrospective study has revealed that cancer patients receiving radiofrequency ablation prior to neoantigen vaccination showed better survival [43]. The murine model used in this study further validated the synergistic effect of neoantigen vaccination, ICIs and radiofrequency ablation.

Together, these studies have highlighted the promise of combinational approaches in overcoming immune exhaustion and broadening PCV-induced antitumor responses. Further investigations in well-defined patient populations are warranted to clarify how to best integrate PCVs with other modalities to achieve enhanced and durable clinical outcomes in cancer patients.

## 8. Trends and Future Perspectives

Over the past decade, PCVs have progressed from proof-of-concept immunogenicity studies [9,10,63] toward more advanced clinical testing and combinational strategies [15,66,67,72]. Notably, both BioNTech and Moderna have advanced their PCV candidates (**Autogene cevumeran** and **mRNA 4157**) into large Phase II trials, with Moderna leading in the field in early Phase III studies. Previously, Moderna’s **mRNA-4157** in combination with pembrolizumab achieved a statistically significant improvement in RFS over pembrolizumab alone in resected high-risk melanoma, reducing the risk of recurrence or death by around 44%. This secured **mRNA-4157** a Breakthrough Therapy Designation from both the Food and Drug Administration (FDA) and European Medicines Agency (EMA). Currently, Moderna’s platform is broadly developed by combining **mRNA-4157** with pembrolizumab.

PCV production is constrained by multiple challenges, including the need for rapid sequencing, individualized neoantigen prioritization, and manufacturing of customized GMP-grade vaccines within a feasible clinical window. Technological barriers include variable quality control pipelines and limited automation in generating neoantigen products. Regulatory challenges arise because each vaccine batch is unique, requiring individualized release criteria, sterility testing, and chain-of-identity tracking. Modular GMP manufacturing units equipped with automated peptide synthesizers or microfluidic-based mRNA production facilities have managed to reduce turnaround times. To further improve scalability of PCV, AI-driven workflow optimization, standardized QC frameworks and centralized regulatory pathways for individualized biologics are expected to be employed in the near future.

Concurrently, the clinical applications of PCVs are diversifying. Many ongoing and planned studies now position PCVs as adjuvant therapies, aiming to reduce tumor recurrence after surgical resection, radiotherapy, or other tumor-debulking interventions. The success of **mRNA-4157** in combination with pembrolizumab has underscored the trend of combinational therapy for disease maintenance, supporting the rationale of using such combination in patients in an earlier disease setting. A smaller but growing subset of trials is now exploring PCVs in first-line settings in advanced diseases, though these tend to remain in early-phase (Phase 1/2) designs and may face challenges in recruiting patients with less treatment experience. Employing PCVs in earlier lines of therapy is not yet routine in most clinical settings due mainly to logistical, regulatory, and risk-benefit considerations; however, such studies are important to further unleash the clinical potential of PCVs.

To enhance the clinical efficacy of PCVs, researchers have been exploring complementary strategies. One such strategy is rational antigen selection and immunogenicity validation throughout vaccination to dynamically optimize the vaccine formulation. For example, Oyama et al. used a two-stage process where patient-specific autologous DCs were pulsed with neoantigen peptides, and ex vivo ELISpot was then used to identify responsive epitopes [53]. Unresponsive epitopes were eliminated from the neoantigen peptide pool to stimulate DCs for a second round of vaccination. This approach helps enrich for epitopes that trigger T-cell activation rather than solely relying on in silico predictions. Another underexplored but intriguing strategy is dynamic or serial batches of PCVs, that is, continuously updating the vaccine neoantigen panel during therapy to account for evolving tumor mutations and clonal drift, although such strategy can be limited by low mutational burden of certain tumors.

Further, some route of administration innovations may improve vaccine potency by engaging more APCs. Mixed or “hybrid” routes could spatially diversify antigen uptake and presentation, possibly broadening the T-cell repertoire. Similarly, adjusting prime-boost regiments is a key variable. How frequently and how closely to separate priming doses, and how to time subsequent boosting remain an open area for optimization.

Off-the-shelf shared neoantigen vaccines are emerging as a strategy for patients with common mutations. These vaccines, directed at recurrent oncogenic mutations (e.g., KRAS, TP53, or IDH1), have the potential to be deployed while PCVs are being manufactured. A hybrid model combing shared neoantigen vaccines and PCVs may prevail, with initial shared neoantigen vaccines given as prime, followed by PCVs as boosters. Meanwhile, accelerating sequencing pipelines and employing AI for neoepitope prediction may shorten the design-to-dose interval.

Because tumors are heterogenous and adaptive, combinational therapies are likely essential for durable success. Other than ICIs, in situ vaccination priming strategies using ablation (radiofrequency or cryoablation) may have the potential to improve the clinical outcome of PCVs [43,85,86]. Ablative modalities may promote antigen release and local inflammation, priming the immune system with neoantigens. PCVs can then enrich and amplify neoantigen-specific immune responses, assisting the expansion of neoantigen-reactive T cells. Additionally, other in situ vaccination strategies, such as the use of nanorods to help generate neoantigen-carrying autophagosomes for the presentation to APCs, have shown promises in mounting neoantigen-specific responses in mice [87,88]. Combining these strategies with PCVs may show synergistic effect to generate robust antitumor immune responses.

A central limitation in current vaccine trials is the limited options of clinically approved adjuvants. Whereas mRNA vaccines carry intrinsic adjuvanticity and utilize LNPs for better APC recognition and uptake, peptide PCVs require exogenous adjuvants to trigger APC activation. Yet only a few immune-stimulating agents such as poly-ICLC have been validated in oncology, and head-to-head comparisons are lacking. To fully exploit synergistic adjuvanticity, randomized studies comparing adjuvant classes such as TLR agonists and STING agonists are needed to define optimal adjuvant pairings in PCVs.

On the immunobiological front, the field is just beginning to dissect the diversity of tumor-infiltrating T-cell phenotypes beyond “bulk responders”. Emerging evidence has underscored the pivotal role of CD4^+^ neoantigen-reactive T cells, which not only secrete proinflammatory cytokines such as IFN-γ and TNF, but can directly kill tumor cells via Fas-FasL interactions [89], license DCs through CD40/CD40L engagement to support CD8^+^ priming [90], and mitigate CD8^+^ exhaustion [91]. Intriguingly, studies have suggested that some tumors downregulate MHC class I but retain class II expression, making them vulnerable to CD4^+^ T cell-mediated control. Currently, allele-specific MHC-II ligands prediction tools such as MixMHC2pred [92] and NeonMHC2 [93], have been made available and incorporated in multi-parameter prioritization frameworks to optimize CD4^+^ epitope inclusion. Although many studies profiled memory versus effector subsets and measured the production of proinflammatory cytokines such as IFN-γ and TNF and anti-inflammatory cytokines such as IL-10, fewer have studied unconventional CD4^+^ T cell populations such as Th9 cells which have been reported to have promising antitumor activity [94,95,96,97] or rare CD4^+^CD8^+^ double-positive T cells that may constitute a large proportion of the tumor-infiltrating lymphocytes [98,99]. Whether these CD4^+^ T cell subsets play immunoregulatory or antitumor roles in the neoantigen setting requires further studies.

Another evolving understanding of tumor-infiltrating lymphocytes involves T-cell avidity (i.e., the strength of TCR-pMHC interactions). Counterintuitively, recent work indicated that low-avidity T cells may outperform high-avidity clones in long-term tumor control [100]. Singhaviranon et al. showed that low-avidity neoantigen-specific CD8+ T cells were more effective and responsive to ICI, whereas high-avidity clones exhibited transcriptomic exhaustion and limited therapeutic control. In both mice and human, low-avidity T cells were found to be the principal effectors to suppress tumor growth. These findings suggested that neoantigen PCVs may benefit from selecting suboptimal affinity neoantigens that avoid premature exhaustion. Interestingly, earlier work indicated that low-avidity clones may not hinder high avidity T cell responses [101]. Inclusion of some subdominant epitopes in PCVs may preferentially expand the resilient clones, contributing to long-lasting antitumor responses through expanding the resilient clones. Understanding how to sustain low-avidity clones and prevent activation-induced cell death may therefore be critical for long-term vaccine efficacy.

Another overlooked dimension within the immunology space of neoantigen vaccination is humoral immunity. Although only a few neoantigen vaccine studies have reported neoantigen-specific IgG responses post vaccination [65], whether such responses contribute to tumor opsonization, complement activation, or antibody-dependent cellular cytotoxicity remains unclear. In addition, induction of neoantigen-specific IgE or allergic reactions have not been systemically studied. Bridging such gaps of knowledge will help implement caution and monitoring for future trials.

## 9. Conclusions

In summary, despite rapid advances of PCVs in the past decade, a number of translational and operational bottlenecks still remain. First, the pipeline from tumor sequencing to vaccine formulation is labor intensive, as it relies heavily on in silico neoantigen prediction with imperfect accuracy, and validation of TCR reactivity is barely feasible at scale. Second, the manufacturing window ranging from 6 to 16 weeks limits the applicability of PCVs in fast-progressing disease settings. Third, limited trials have incorporated comparisons across vaccine adjuvant types, immunization routes, or prime-boost schedules, making it difficult to ascribe observed effects to design variables. Lastly, most clinical studies categorized tumor-infiltrating lymphocytes at a low resolution and overlooked the role of neoantigen-specific humoral immune responses, limiting mechanistic insights. By integrating intelligent antigen selection, refined delivery and scheduling, novel adjuvants, biomarker-driven patient stratification, as well as multimodal treatment synergy, PCVs may show the potential to fulfill the promise of inducing long-term antitumor immune responses to control tumors. The next generation of PCV clinical trials that are more randomized and more mechanistically informed will determine whether neoantigen vaccination can ascend from niche intervention to a mainstream cancer therapy.

## Figures and Tables

**Figure 1 vaccines-13-01231-f001:**
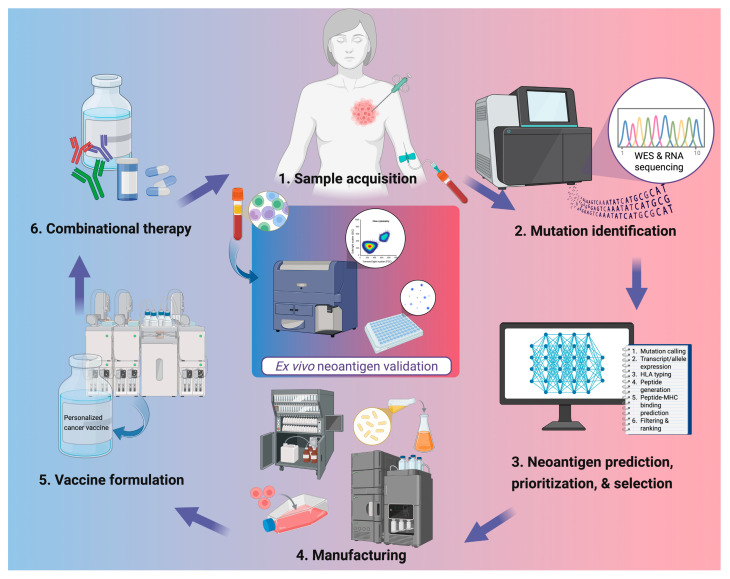
Workflow for personalized neoantigen-based cancer vaccine development. Tumor tissue and peripheral blood are collected for genomic analysis, followed by whole-exome and RNA sequencing to identify somatic mutations. Candidate neoantigens are predicted, prioritized, and selected using computational pipelines integrating parameters such as mutation calling, HLA typing, and peptide–MHC binding predictions. Validated targets are then manufactured into vaccine products and formulated with appropriate delivery systems or adjuvants. In clinics, PCVs are administered alone or alongside other therapies such as ICIs or chemotherapies. Ex vivo validation of neoantigen-specific immune responses is often done through flow cytometric analysis or IFN-γ ELISpot using patient peripheral blood mononuclear cells. Created in BioRender. Lu, L. (2025) https://BioRender.com/d0z6isr.

**Figure 2 vaccines-13-01231-f002:**
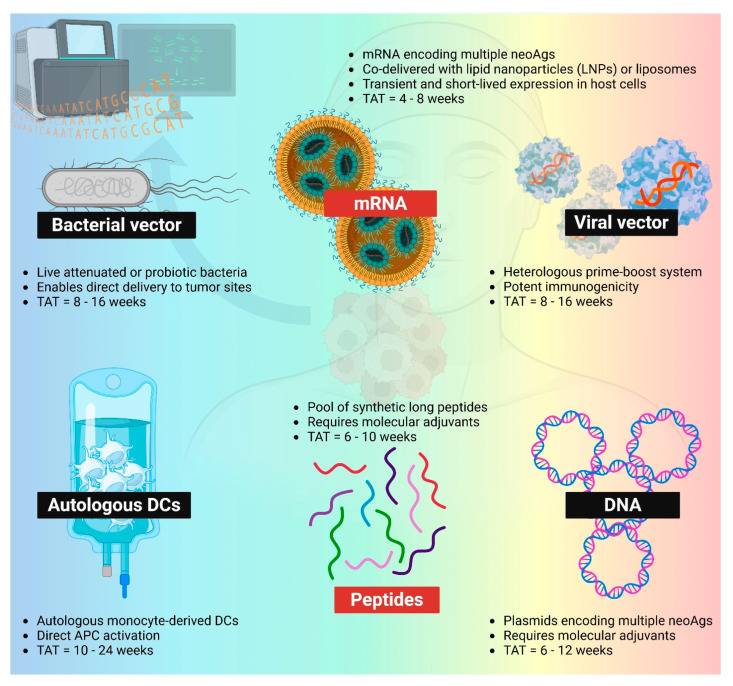
Different PCV platforms. mRNA (LNP-mRNA or mRNA-lipoplex) and peptides remain the most popular two platforms within the space of PCVs. Bacterial vector, viral vector, autologous DCs, and DNA neoantigen vaccines are also being evaluated in clinics. TAT: turnaround time. NeoAgs: neoantigens. APC: antigen-presenting cells. Created in BioRender. Lu, L. (2025) https://BioRender.com/uxoiblr.

**Table 1 vaccines-13-01231-t001:** Comparison of Different PCV Platforms.

PCV Platforms	Advantages	Disadvantages	Bottlenecks
Peptide	• Can be lyophilized; does not require cold-chain transportation • Straightforward GMP synthesis	• Low immunogenicity; requires immune adjuvants to boost the immune response	• Lack of robust immune adjuvants that are commercially available
mRNA	• Easy to encode multiple antigens in a single construct • Highly scalable • Fastest manufacturing time	• Requires liposome or lipid nanoparticles for optimized delivery • Requires cold-chain logistics	• Variation in LNP encapsulation efficiency
Plasmid DNA	• Self-adjuvanted • Easy to encode multiple antigens in a single construct • Cheap and scalable production	• Weaker immunogenicity compared to mRNA • Requires electroporation for efficient cellular uptake	• Low transfection efficiency
Pulsed-DC	• No need for in vivo antigen uptake • Direct activation of APC ex vivo • Tailored antigen loading • Highly potent T cell priming	• Longer manufacturing time • Expensive and involves complicated cell culturing • Variable DC phenotypes from different patients	• Batch-to-batch inconsistency in DC phenotypes and viability • Regulatory complexity • Poor scalability in large trials
Viral Vector	• Highly immunogenic and self-adjuvanted • Efficient antigen delivery • Can induce robust CD8+ T cell responses	• Concerns of redundant immune responses against vectors • Requires a heterologous prime/boost system to avoid excessive immunity against vectors	• Anti-vector immunity
Bacterial Vector	• Highly immunogenic and self-adjuvanted • Can target tumor via intracellular infection	• Safety concerns due to live attenuated bacteria • Risk of systemic inflammation • Complex regulatory pathways	• Limited experience in PCV compared to viral vectors • Manufacturing requires BSL-2+ or BSL-3 controls • Immunodominance toward bacterial antigens over neoepitopes

**Table 2 vaccines-13-01231-t002:** Examples of PCV Clinical trials.

Vaccine Name, Clinical Trial Identifier & Clinical Stage	Platform	Targets	Treatment Details	Clinical Outcome
**PGV-001** [65] NCT02721043 Phase 1	Peptide	Breast, head and neck, lung, multiple myeloma, & urothelial cancer	Post-surgery **adjuvant therapy** Poly-ICLC and tetanus toxoid peptide as adjuvants Additional poly-ICLC given one day post vaccination	Median OS = 51.5 months Median RFS = 49.0 months
**NEO-PV-01** [14] NCT0897765 Phase 1b	Peptide	Advanced melanoma, NSCLC, Bladder cancer	Poly-ICLC as adjuvant In combination with nivolumab	ORR = 59%, 39% and 27% for melanoma, NSCLC, and bladder cancer. Median PFS = 23.5, 8.5, and 5.8 months for melanoma, NSCLC, and bladder cancer. Median OS not reached for melanoma and NSCLC, 20.7 months for bladder cancer.
**NEO-PV-01** [8] NCT03380871 Phase 1b	Peptide	Non-squamous NSCLC	Poly-ICLC as adjuvant In combination with chemotherapy and pembrolizumab as **first line therapy**	ORR = 69% CBR = 100% Median PFS = 7.2 months Median OS = 20 months
**NeoVax** [66] NCT03929029 Phase 1	Peptide	Previously untreated melanoma (Stage IIIB/C/D)	Poly-ICLC and Montanide as vaccine adjuvant Co-administered with ipilimumab Prior line of systemic nivolumab	CD8^+^ T cell responses tested in 6/9 patients. One patient entered the study with unresectable melanoma achieved CR. Five patients entered the study with NED remained NED, three patients had recurrence.
**NeoVax** [67] NCT02950766 Phase 1	Peptide	Renal cell carcinoma (stage III or IV)—low mutational burden	**Adjuvant therapy** In combination w/wo Ipilimumab	No recurrence of RCC at the median follow-up of 40.2 months after surgery.
**iNeo-Vac-P01** [43,68] NCT03662815 Phase 1	Peptide	Advanced malignant solid tumors	GM-CSF as adjuvant, given 30 min prior to vaccination	Disease control rate was 71.4%; median PFS 4.6 months [68]. Patients receiving radiofrequency ablation prior to vaccination displayed longer median PFS and OS (4.42 and 20.18 months) than those who did not (2.82 and 10.94 months) [43].
**iNeo-Vac-P01** [69] NCT03645148 Phase 1	Peptide	Advanced pancreatic cancer refractory to standard treatment	GM-CSF as adjuvant, given 30 min prior to vaccination	Mean OS = 24.1 months/8.3 months (associated with vaccine treatment) Mean PFS = 3.1 months
Neoantigen peptide vaccine NCT03606967 Phase 2	Peptide	Metastatic TNBC	Poly-ICLC as adjuvant	No data reported yet.
Personalized NeoAntigen Peptide Vaccine [70] NCT06314087 Phase 2	Peptide	Refractory solid tumors	In combination with radiotherapy	No data reported yet.
**mRNA 4157** [46] NCT03313778 Phase 1	mRNA	Unresectable metastatic HPV^−^ HNSCC)	In combination with pembrolizumab	Response rate = 27.3% (6/22) Best overall response = 2 CR, 4 PR, and 8 SD Median PFS = 15.0 weeks Median OS = 107.1 weeks
**mRNA 4157** [45] NCT03897881 Phase 2b	mRNA	Resected melanoma	In combination with Pembrolizumab	44% reduction in recurrence when neoantigen vaccine combined with pembrolizumab compared with pembrolizumab monotherapy.
**mRNA 4157** [71] NCT06077760 Phase 3	mRNA	NSCLC	**Adjuvant therapy** In combination with Pembrolizumab	No data reported yet.
**mRNA 4157** NCT05933577 Phase 3	mRNA	High-risk melanoma	**Adjuvant therapy** In combination with Pembrolizumab	No data reported yet.
**Autogene cevumeran** [72] NCT04161755 Phase 1	mRNA	Surgically resected pancreatic ductal adenocarcinoma	**Adjuvant therapy** In combination with atezolizumab and 4-drug chemotherapy	Responders had longer median recurrence-free survival (not reached) than non-responders (13.4 months) at 18-month median follow up. Objective response rate = 30.6% Complete response = 8.3%.
**Autogene cevumeran** [15] NCT03289962 Phase 1	mRNA	Advanced solid tumors	Monotherapy or in combination with atezolizumab in pretreated patients	One CR reported in the monotherapy cohort, and two in the combination cohort. ORR in ICI naïve cohort: melanoma and renal cell cancer = 33.3%, urothelial cancer = 18.2%, NSCLC = 10.0%, and TNBC = 0%
**Autogene cevumeran** [73] NCT04486378 Phase 2	mRNA	ctDNA positive, surgically resected stage II/III rectal cancer or stage II (high risk)/stage III colon cancer	**Adjuvant therapy** Monotherapy	No data reported yet.
**Autogene cevumeran** NCT05968326 Phase 2	mRNA	Resected pancreatic ductal adenocarcinoma	**Adjuvant therapy** Monotherapy of mFOLFIRINOX vs. Autogene cevumeran + atezolizumab + mFOLFIRINOX	No data reported yet.
**GNOS-PV02** [51] NCT04251117 Phase 1/2	Plasmid DNA	Advanced hepatocellular carcinoma	In combination with pembrolizumab and plasmid IL-12	ORR = 30.6%; DCR = 55.6% Median PFS = 4.2 months Median OS = 19.9 months DoR = not reached
**Personalized Polyepitope DNA Vaccine** [37] NCT02348320 Phase 1	Plasmid DNA	TNBC	**Adjuvant therapy**	RFS was 87.5% compared to historical data 49% after 36 months of follow-up.
**Personalized Polyepitope DNA Vaccine** NCT03199040 Phase 1	Plasmid DNA	TNBC	**Adjuvant therapy** W/wo durvalumab	Study terminated.
Autologous Lymphoma Immunoglobulin-derived scFv-chemokine DNA Vaccine NCT01209871 [49] Phase 1	Plasmid DNA	Lymphoplasmacytic lymphoma	**First line therapy** (in untreated patients) LPL-derived Ig single chain variable fragment (ScFv) fused to chemokine CCL20 in DNA plasmid.	One patient achieved a minor response, and eight patients had stable disease.
**VB10.NEO** [74] NCT05018273 Phase 1b	Plasmid DNA	Pan-cancer (locally advanced and metastatic tumors such as melanoma, NSCLC, CRCC, bladder cancer, and HNSCC	In combination with atezolizumab	Responses against 53% of neoepitopes on average were reported [74].
**EVX-02** [75] NCT0445503 Phase 1/2	Plasmid DNA	Late staged melanoma (Stage III or IV)	In combination with nivolumab	N/A
Mature dendritic cell vaccine [63] NCT00683670 Phase 1	Autologous DCs	Advanced melanoma (unresectable stage III or IV melanoma)	In combination with chemotherapy DC pulsed with two gp100 peptides and 10 neoantigen peptides	All three patients generated neoantigen-specific CD8^+^ T cell responses.
**Neo-DCVac** [52] NCT02956551 Phase 1	Autologous DCs	Refractory NSCLC	DC loaded with peptide neoantigens; GM-CSF administered as adjuvant	Objective effectiveness rate = 25% Disease control rate = 75% Median PFS = 5.5 months Median OS = 7.9 months
**Neo-mDC** [42] NCT04078269 Phase 1	Autologous DCs	Resected NSCLC	**Adjuvant therapy** DC loaded with mRNA neoantigens	Recurrence free remained in 8 out of 9 patients during a median follow-up of about 30 months.
**nDC** [54] NCT04968366 Phase 1	Autologous DCs	Glioblastoma Multiforme	**Adjuvant therapy** after radiotherapy and chemotherapy DC loaded with peptide neoantigens	12-month PFS = 87.5% 12-month OS = 100%
**Neo-P DC** [53,55] Phase 1	Autologous DCs	Postoperative pancreatic cancer	**Adjuvant therapy** Postoperative recurrence cases and adjuvant setting	In recurrence cases, responders showed longer OS than non-responders.
**ADXS-NEO** [59] NCT03265080 Phase 1	Bacterial vector (listeria monocytogenes)	Colorectal carcinoma, Head and neck squamous cell carcinoma, NSCLC	Listeriolysin O incorporated with neoantigens as adjuvant In combination with pembrolizumab	Study terminated.
**NOUS-PEV** [61] NCT04990479 Phase 1	Viral vector	Metastatic melanoma	Great ape adenovirus as prime, Modified Vaccina Virus Ankara as boost In combination with pembrolizumab as **first line therapy**	Survival data not reported; Best overall response: 16.67% (1/6) CR, 50.00% (3/6) PR, 16.67% (1/6) SD, and 16.67% (1/6) PD.
**GRANITE** [62] NCT05141732 Phase 2	Viral vector	Newly diagnosed metastatic microsatellite stable colorectal cancer	Chimpanzee adenovirus as prime, self-amplifying mRNA as boost In combination with ICI + first line standard of care	Risk of disease progression or death in the GRANITE group was 27% lower than that in the control group (patients receiving first line standard of care only).

ORR: objective response rate; DCR: disease control rate; CBR: clinical benefit rate; DoR: median duration of response; OS: overall survival; PFS: progression-free survival; RFS: recurrence free survival; NED: no evidence of disease; CR: complete response; PR: partial response; SD: stable disease; PD: progression disease; TNBC: triple-negative breast cancer; NSCLC: non-small cell lung cancer.

## Data Availability

No new data were created or analyzed in this study.

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
