# Peer review of "Personalized Cancer Vaccines: Current Advances and Emerging Horizons"

_vaccines, 2025, doi:10.3390/vaccines13121231_

Round 1

Reviewer 1 Report

Comments and Suggestions for Authors

The manuscript analyzes personalized cancer vaccines (PCV), focusing on neoantigens, clinical trials based on PCV, adjuvants, and combination therapy.

The manuscript is concise and well written. The presentation is straightforward and clear.

I would add a table indicating the pros, cons, and bottlenecks of the generation of PCV.

Author Response

We greatly appreciate your kind words to our review. We have added a new table discussing the advantages, disadvantages and bottlenecks of different PCV platforms. Please see new Table 1.

Reviewer 2 Report

Comments and Suggestions for Authors

Dear Editor,

  This review focuses on neoantigen-based personalized cancer vaccines, a revolutionary oncology frontier using individual tumor genetics for targeted immunity. It overviews platforms like peptide, mRNA, and their future, discusses combining adjuvants, delivery systems, and immune checkpoint inhibitions to address immune exhaustion. Despite clinical progress, challenges in neoantigen prediction and personalized manufacturing remain. It highlights advances, bottlenecks, and the need for biomarker-guided trials to unlock their clinical potential in precision immuno-oncology.

  However, this work contains evident logical deficiencies. It requires major revisions before publication. There are several questions as follows:

  1. The overall manuscript structure suffers from logical disconnections and content redundancy between key sections. For example, "Identification of Neoantigens" and "Neoantigens – The Game Changer" cover similar ground about neoantigen sources and importance.
  2. The manuscript contains only one figure. It lacks schematic diagrams for key processes. Examples include the neoantigen selection pipeline, vaccine mechanism of action, or combination therapy strategies. This affects readability and information delivery.
  3. The "personalized" aspect in the title is not fully developed in the text. The manuscript lacks a systematic description of key steps for personalized vaccine creation. Additionally, the impact of individual differences on vaccine efficacy is also not deeply discussed.
  4. The manuscript's structure is unbalanced. The introduction provides insufficient background information, while section 8 "Trends and Future Perspectives" contains excessive content.
  5. The keywords in this review are too numerous. Please simplify them.
  6. This review contains some repetitive sentences. The sentences in lines 47-48 are identical to those in lines 46-47.
  7. There are problems with academic rules in the text. These mistakes need to be fixed following academic practices.
    • All tables must adopt the standard three-line format, in accordance with academic convention, to ensure clarity and professionalism.
    • Some CD4+ and CD8+ lack the superscript symbol.
    • It contains non-standard academic term formats, including "in vivo" presented in plain text instead of italics. These inconsistencies need to be corrected in accordance with academic conventions.
    • Table 1 has inconsistent font sizes. The characters in the first row are larger than those in the other rows.
    • Reference format of this review is inconsistent. For example, the number of authors is not uniform. Please make the correction.

Author Response

  1. The overall manuscript structure suffers from logical disconnections and content redundancy between key sections. For example, "Identification of Neoantigens" and "Neoantigens – The Game Changer" cover similar ground about neoantigen sources and importance.

Thank you for your thoughtful comments. We respectfully disagree with the reviewer’s specific example. Section 2 discusses the biological significance, immunological relevance, and clinical role of neoantigens, whereas Section 3 focuses on methodological aspects of sequencing-based neoantigen identification and computational prediction tools. These topics are scientifically distinct and serve different pedagogical purposes.

  1. The manuscript contains only one figure. It lacks schematic diagrams for key processes. Examples include the neoantigen selection pipeline, vaccine mechanism of action, or combination therapy strategies. This affects readability and information delivery.

Thank you for your insightful comment. We have added a new figure in the manuscript (please see question #3).

  1. The "personalized" aspect in the title is not fully developed in the text. The manuscript lacks a systematic description of key steps for personalized vaccine creation. Additionally, the impact of individual differences on vaccine efficacy is also not deeply discussed.

Thank you for this insightful comment. The manuscript indeed discusses the principles of personalization throughout (e.g., tumor-specific mutation profiles, HLA diversity, de novo T-cell priming, and patient-specific neoantigen selection), but we agree that the workflow for personalized vaccine generation was not presented as a consolidated, stepwise process.

To address this, we have added a new figure that systematically outlines the key steps from tumor sampling, sequencing, neoantigen prediction, prioritization, validation, to manufacturing and clinical implementation.

These additions strengthen the emphasis on the personalized nature of PCVs and improve readability for non-specialist audiences.

  1. The manuscript's structure is unbalanced. The introduction provides insufficient background information, while section 8 "Trends and Future Perspectives" contains excessive content.

Thank you for this comment. To address this concern, we have expanded the Introduction to provide clearer background and context for readers. Regarding Section 8, we respectfully note that future perspectives are a central component of review articles, especially in rapidly developing fields. This section synthesizes emerging concepts and ongoing translational challenges and therefore requires a comprehensive discussion. With the expanded introduction, the manuscript is now more structurally balanced while preserving the depth of Section 8. We appreciate the reviewer’s insight and believe the revisions significantly strengthen the manuscript.

  1. The keywords in this review are too numerous. Please simplify them.

Thank you for your suggestion. We have simplified the keywords to cover only neoantigens, personalized cancer vaccines, peptide vaccines, genetic vaccines, autologous DC vaccines, and vector vaccines.

  1. This review contains some repetitive sentences. The sentences in lines 47-48 are identical to those in lines 46-47.

Thank you for your careful review. We have removed lines 47-48.

  1. There are problems with academic rules in the text. These mistakes need to be fixed following academic practices.

Thank you for noting these issues. We performed a comprehensive, line-by-line revision of the manuscript to correct all deviations from academic writing conventions. Specifically, we have:

  • Standardized scientific terminology throughout the manuscript.
  • Corrected formatting of technical expressions (e.g., in vivo, ex vivo, in vitro now in italics).
  • Ensured all cell subset markers use proper superscripts (e.g., CD4⁺, CD8⁺, IFN-γ⁺).
  • Revised inconsistent capitalization, hyphenation, and abbreviation usage.
  • Improved grammar, sentence structure, and eliminated minor stylistic errors.
  • Reformatted Table 1 according to journal academic formatting guidelines.
  • Ensured consistent reference formatting across all citations.

  1. All tables must adopt the standard three-line format, in accordance with academic convention, to ensure clarity and professionalism.

Thank you for your thorough review. We have adopted three-line format.

  1. Some CD4+ and CD8+ lack the superscript symbol.

Thank you for your careful review. We have added superscript symbol. Please see L228, 229 and 397.

  1. It contains non-standard academic term formats, including "in vivo" presented in plain text instead of italics. These inconsistencies need to be corrected in accordance with academic conventions.

Thank you for your observations. We have corrected these accordingly (please see question #7).

  1. Table 1 has inconsistent font sizes. The characters in the first row are larger than those in the other rows.

Thank you for your careful review. The first row is the title for the subsections of the following content, but we have made the font size consistent according to your suggestion.

  1. Reference format of this review is inconsistent. For example, the number of authors is not uniform. Please make the correction.

Thank you for your careful review. We have applied MDPI referencing style. All authors’ names are spelled out.

Reviewer 3 Report

Comments and Suggestions for Authors

This is a comprehensive review, well written but still needing some modification as summarized. 

  1. The abstract emphasizes significant obstacles in the prediction of immunogenic neoantigens. Could the authors elucidate the bioinformatic or experimental limitations that most significantly impact the precision of neoantigen detection and how recent AI or multi-omics methodologies are mitigating these deficiencies? Please provide a brief summary of the information presented in the abstract.

  1. Considering the focus on personalized vaccination platforms, what are the primary technological and regulatory challenges in scaling individualized vaccine production, and how could standardized processes enhance clinical viability? Describe in detail in the revised MS.

  1. The review discusses the integration of vaccinations with immune checkpoint inhibition to mitigate immunological exhaustion. Could the authors elucidate specific mechanistic or clinical data that substantiates this synergy, along with any hazards or unfavorable immunological effects? Please explain it.

  1. The section presents promising shared neoantigen vaccines while also highlighting problems with immune dilution caused by non-patient-specific epitopes. Could the authors elucidate whether there exists comparison evidence or a mechanistic rationale that illustrates the performance of shared neoantigen vaccines in relation to fully individualized vaccinations concerning T-cell clonality, immunogenicity, and clinical efficacy? Update this is the revised MS.

  1. The article emphasizes the prevalence of neoantigen-reactive CD4⁺ T cell responses, even if epitope selection is frequently dependent on HLA class I. Could the authors elaborate on how vaccine design ought to evolve to more effectively leverage CD4⁺ T cell-mediated antitumor immunity and whether MHC-II-focused neoantigen prediction algorithms are being successfully incorporated into existing frameworks? Explain in detail.

  1. The manuscript emphasizes shared neoantigen vaccines; however, could the authors elucidate the immunological trade-offs between shared and individualized techniques for immune specificity, TCR repertoire diversity, and the potential for tumor evasion?

  1. The discussion regarding the predominance of CD4⁺ T cells in neoantigen responses is noteworthy. Could the authors include additional mechanistic insights or recent information about the specific roles of CD4⁺ subsets (Th1, Th9, Tfh) in tumor immunity?

  1. The discussion of false-positive rates in neoantigen prediction appears comprehensive. Could the authors provide further details on experimental approaches to validate immunogenicity (e.g., mass spectrometry, single-cell TCR sequencing, and organoid systems)?

  1. Could the authors elaborate on the regulatory, logistical, and manufacturing challenges related to scaling personalized vaccine production and how automation, or artificial intelligence, can mitigate turnaround times?

  1. The authors describe "low-avidity T cells" as beneficial. Could they clarify how vaccine design may selectively promote or maintain these clones to avert exhaustion while ensuring efficacy?

Author Response

  1. The abstract emphasizes significant obstacles in the prediction of immunogenic neoantigens. Could the authors elucidate the bioinformatic or experimental limitations that most significantly impact the precision of neoantigen detection and how recent AI or multi-omics methodologies are mitigating these deficiencies? Please provide a brief summary of the information presented in the abstract.

Thank you for this important suggestion. In the revised manuscript, we have expanded the discussion in section 3.

“The major limitations in neoantigen prediction stem from inaccurate modeling of pep-tide-MHC binding, incomplete characterization of antigen processing, insufficient integration of post-translational events, and the inability to predict TCR recognition. Many current pipelines rely on neural networks trained on limited datasets of eluted ligands, overlooking the diversity of HLA alleles or proteasomal cleavage rules. Recent models using artificial intelligence (AI) models such as transformer-based architectures trained on numerous peptide-HLA interaction measurements, improve prediction of stability rather than affinity alone [33]. Additionally, multi-omics workflows that integrate genomics, transcriptomics and immunopeptidomics such as ScanNeo2 were reported to reduce false-positive rates by accounting for RNA expression, allele-specific presentation, and neoepitope abundance [31,32].”

  1. Considering the focus on personalized vaccination platforms, what are the primary technological and regulatory challenges in scaling individualized vaccine production, and how could standardized processes enhance clinical viability? Describe in detail in the revised MS.

Thank you for this valuable suggestion. We have added a focused discussion on technological, logistical, and regulatory barriers to scalable personalized vaccine production, including automation and standardization strategies (please see section 8. Trends and Future Perspectives).

“PCV production is constrained by multiple challenges, including the need for rapid sequencing, individualized neoantigen prioritization, and manufacturing of customized GMP-grade vaccines within a feasible clinical window. Technological barriers include variable quality control pipelines and limited automation in generating neoantigen products. Regulatory challenges arise because each vaccine batch is unique, re-quiring individualized release criteria, sterility testing, and chain-of-identity tracking. Modular GMP manufacturing units equipped with automated peptide synthesizers or microfluidic-based mRNA production facilities have managed to reduce turnaround times. To further improve scalability of PCV, AI-driven workflow optimization, standardized QC frameworks and centralized regulatory pathways for individualized bio-logics are expected to be employed in the near future.”

  1. The review discusses the integration of vaccinations with immune checkpoint inhibition to mitigate immunological exhaustion. Could the authors elucidate specific mechanistic or clinical data that substantiates this synergy, along with any hazards or unfavorable immunological effects? Please explain it.

Thank you for this comment. It is widely believed that ICIs could support the adequate effector functions of TILs including infiltrative neoantigen-reactive T cells. Hence, we have indicated in section 7 “This approach helps re-invigorate tumor-infiltrative cytotoxic T cells and maintains their effector functions where immunosuppression presents.”

We have cited an animal work that addressed this question (please see section 7). “Combinational use of ICIs and neoantigen vaccines in murine models have shown synergistic effects in promoting higher infiltration of CD8+ T resident memory cells into tumors, contributing to stronger antitumor responses [82].”

Regarding to the hazards or unfavorable immunological effects, we were unable to find any noteworthy clinical side effects using this combinational approach, as side effects such as colitis or dermatitis using this approach are similar to ICI monotherapy. Hence, we decided not to mention it in the manuscript.

  1. The section presents promising shared neoantigen vaccines while also highlighting problems with immune dilution caused by non-patient-specific epitopes. Could the authors elucidate whether there exists comparison evidence or a mechanistic rationale that illustrates the performance of shared neoantigen vaccines in relation to fully individualized vaccinations concerning T-cell clonality, immunogenicity, and clinical efficacy? Update this is the revised MS.

Head-to-head studies discerning the performance difference between shared neoantigen vaccines and PCVs are lacking in human studies. We cited an animal work that compared fully personalized vaccine with shared neoantigen vaccine in a murine 4T1 model. Please see section 2. “Peterson et al. compared personal and shared frameshift neoantigen vaccines in a mouse breast cancer model [23]. Both personal and shared neoantigen vaccines elicited robust neoantigen-specific T cell responses, showing comparable protection against tumor growth. This suggested that when high-quality shared neoantigens exist, a shared vaccine has the potential to match a personalized vaccine in generating anti-tumor responses.”

  1. The article emphasizes the prevalence of neoantigen-reactive CD4⁺ T cell responses, even if epitope selection is frequently dependent on HLA class I. Could the authors elaborate on how vaccine design ought to evolve to more effectively leverage CD4⁺ T cell-mediated antitumor immunity and whether MHC-II-focused neoantigen prediction algorithms are being successfully incorporated into existing frameworks? Explain in detail.

Thank you for this insightful suggestion. We have included such discussion in the future perspective section. Please see below.

“Currently, allele-specific MHC-II ligands prediction tools such as MixMHC2pred [92] and NeonMHC2 [93], have been made available and incorporated in multi-parameter prioritization frameworks to optimize CD4+ epitope inclusion.”

  1. The manuscript emphasizes shared neoantigen vaccines; however, could the authors elucidate the immunological trade-offs between shared and individualized techniques for immune specificity, TCR repertoire diversity, and the potential for tumor evasion?

Thank you for this insightful comment. We have included relevant discussion in the last paragraph of section 2. “Shared neoantigens simplify manufacturing but risk lower specificity and higher potential for tumor immune escape. Individualized vaccines incorporating multiple patient-specific neoantigens maximize precision and TCR repertoire depth but require longer production timelines. The choice between these strategies depends on tumor type, mutational burden, and logistical constraints.”

  1. The discussion regarding the predominance of CD4⁺ T cells in neoantigen responses is noteworthy. Could the authors include additional mechanistic insights or recent information about the specific roles of CD4⁺ subsets (Th1, Th9, Tfh) in tumor immunity?

Thank you very much for this suggestion. As this review focuses on neoantigen vaccines, a detailed characterization of Th subsets would go beyond our scope, especially when their neoantigen-reactivity is unknown.

  1. The discussion of false-positive rates in neoantigen prediction appears comprehensive. Could the authors provide further details on experimental approaches to validate immunogenicity (e.g., mass spectrometry, single-cell TCR sequencing, and organoid systems)?

Thank you for bringing this. This is normally done by intracellular cytokine staining (focusing on Th1 cytokines such as INF-g and/or TNF), or IFN-g ELISpot assay. We have modified the paragraph in section 3 to indicate the techniques used. For example, “Zhang et al. found that only 31.1% (14 of 45) of selected neoantigens trigger Th1 cytokine secretion detected by flow cytometry [37].”

  1. Could the authors elaborate on the regulatory, logistical, and manufacturing challenges related to scaling personalized vaccine production and how automation, or artificial intelligence, can mitigate turnaround times?

Thank you for your comment on this. We have briefly discussed these in section 8.

“PCV production is constrained by multiple challenges, including the need for rapid sequencing, individualized neoantigen prioritization, and manufacturing of customized GMP-grade vaccines within a feasible clinical window. Technological barriers include variable quality control pipelines and limited automation in generating neoantigen products. Regulatory challenges arise because each vaccine batch is unique, re-quiring individualized release criteria, sterility testing, and chain-of-identity tracking. Modular GMP manufacturing units equipped with automated peptide synthesizers or microfluidic-based mRNA production facilities have managed to reduce turnaround times. To further improve scalability of PCV, AI-driven workflow optimization, standardized QC frameworks and centralized regulatory pathways for individualized biologics are expected to be employed in the near future.”

  1. The authors describe "low-avidity T cells" as beneficial. Could they clarify how vaccine design may selectively promote or maintain these clones to avert exhaustion while ensuring efficacy?

Thank you for this comment. We have actually already indicated in the paragraph. “These finding suggested that neoantigen PCVs may benefit from selecting suboptimal affinity neoantigens that avoid premature exhaustion.”

To make it more clear, we have added “Inclusion of some subdominant epitopes in PCVs may preferentially expand the resilient clones, contributing to long-lasting antitumor responses through expanding the resilient clones.”

Reviewer 4 Report

Comments and Suggestions for Authors

This review provides a comprehensive and timely overview of the field of neoantigen-based personalized cancer vaccines (PCVs). The authors summarize the journey from neoantigen discovery and prediction to the various vaccine platforms (peptide, mRNA, DNA, dendritic cell, viral, and bacterial vectors), their clinical applications, and the critical roles of immune adjuvants and combinational therapies. The manuscript's main strengths lie in its up-to-date coverage of recent clinical trial data, its clear explanation of the technological and immunological underpinnings of PCVs, and its insightful discussion of current challenges and future perspectives. The inclusion of a detailed table summarizing clinical trials is particularly valuable.
 The review is highly relevant and addresses a rapidly evolving area in precision immuno-oncology. It is comprehensive in its scope, covering the entire pipeline from neoantigen biology to clinical implementation. The identification of key gaps—such as the suboptimal accuracy of in silico prediction tools, the long manufacturing turnaround times, the limited head-to-head comparisons of adjuvants and platforms, and the underexplored role of humoral immunity—is accurate and well-articulated. This makes the review a significant contribution to the field.
The cited references are predominantly recent (within the last 5 years) and highly relevant, including key 2024 and 2025 publications that demonstrate the authors are at the forefront of the field. The references appropriately support the statements made throughout the text. There is no apparent excessive self-citation.
The narrative is logically structured and coherent. Statements and conclusions are, for the most part, well-supported by the provided citations. The discussion on the dominance and multifaceted roles of CD4+ T cells (Sections 2 and 8) is a highlight, reflecting a nuanced understanding of modern tumor immunology beyond a simple CD8+ centric view.

Author Response

Thank you for speaking highly of our review, and we appreciate your help in reviewing it. We have incorporated changes suggested by other reviewers.
